# Highly Effective Methods of Obtaining N-Doped Graphene by Gamma Irradiation

**DOI:** 10.3390/ma13214975

**Published:** 2020-11-05

**Authors:** Piotr Kamedulski, Stanislaw Truszkowski, Jerzy P. Lukaszewicz

**Affiliations:** 1Faculty of Chemistry, Nicolaus Copernicus University, Gagarina 7, 87-100 Torun, Poland; pkamedulski@umk.pl (P.K.); stru@umk.pl (S.T.); 2Centre for Modern Interdisciplinary Technologies, Nicolaus Copernicus University, Wilenska 4, 87-100 Torun, Poland

**Keywords:** carbon materials, N-doped graphene, gamma irradiation, heteroatoms

## Abstract

The design and fabrication of a new effective manufacturing method of heteroatom-doped carbon materials is still ongoing. In this paper, we present alternative and facile methods to obtain N-rich graphene with the use of low energy gamma radiation. This method was used as a pure and facile method for altering the physical and chemical properties of graphene. The obtained materials have an exceptionally high N content—up to 4 wt %. (dry method) and up to 2 wt %. (wet method). High-resolution transmission electron microscopy (HRTEM), scanning electron microscopy (SEM), X-ray diffraction (XRD), Raman spectra and X-ray photoelectron spectroscopy (XPS) studies allowed us to evaluate the quality of the obtained materials. The presented results will provide new insights in designing and optimizing N-doped carbon materials potentially for the development of anode or cathode materials for electrochemical device applications, especially supercapacitors, metal–air batteries and fuel cells. Nitrogen atoms are exclusively bonded as quaternary groups. The method is expandable to the chemical insertion of other heteroatoms to graphene, especially such as sulfur, boron or phosphorus.

## 1. Introduction

Graphene has recently attracted the attention of the research society as well as industry circles. It belongs to the materials category called nano-carbons and particularly to the 2D class of materials. Theoretically, pristine graphene comprises only carbon atoms with some hydrogen atoms at the edge of each graphene plane. The description applies to paternal graphene planes as defined in the materials science encyclopedia and IUPAC documents [1,2]. However, such paternal graphene is hardly applicable in several technological fields despite its useful physicochemical features like chemical stability, excellent electric conductivity and high surface area [3]. Single-layered graphene (SLG) is an expensive material and very difficult to maintain. Therefore, from economic and practical point of view, instead of direct synthesis of SLG and its application, it is better to apply graphite (a 3D structure comprising plenty of parallel oriented and stack SLGs [2]. Utilization of features ascribed to SLG is the primary assumption of such research efforts aiming at graphite splitting to possibly single flakes of graphene. Graphite splitting/exfoliation is not an ideal process and delivers mixed forms regarding the separation degree [4]. Instead of SLG, other forms of graphene are privileged such as few-layered graphene (FLG) and multi-layered graphene (MLG); however, the application potential of these forms is also high [5]. Such forms of exfoliated graphene (SLG, FLG, MLG) may be subjected to further manipulations increasing its applicability in particular fields such as 3D structuring [6,7]. In parallel, graphene might be chemically modified by heteroatom insertion, mainly: oxygen, nitrogen, phosphorus, boron, sulfur, metal/metal oxide clusters. Nitrogen doping belongs to the most popular chemical modifications of graphene [8,9,10]. Generally, N doping is a common strategy for endowing new properties into a material, such as improving conductivity or constructing new bonds with other atoms [11,12,13]. Functional groups, including nitrogen-originated ones play a crucial role in the conversion of the non-polar surface of pristine graphene to the polar one since such introduced heteroatoms like O, S, N have considerably different electronegativity if compared to C atoms. Thus, upon N atom insertion, permanent charges get created on the surface, which are the source of attractive forces between the surface and the molecules (particularly polar) to be adsorbed on the surface. Therefore, N-modified carbon surfaces (as well as O-, S-modified) are naturally predestinated to the deposition of polar molecules and/or ions due to increased intermolecular attraction. The importance of such doping of diversified carbon materials is often presented regarding the application as electrode materials in electrochemical energy generation/storage devices like fuel cell, supercapacitors, metal–air batteries, and Li-ion batteries [14,15,16,17,18]. Additionally, N-modified carbon surfaces (including graphene) may serve as a platform for oligothiophene dyes deposition aiming at the manufacturing of usable photovoltaics materials [19]. Nitrogen-doped carbon materials inclusive of graphene provide high electrical conductivity in comparison to non-enriched ones. The chemically bonded nitrogen atoms play a double role: as an electron donor and are adsorption/catalytic centers of high basic nature (potential donor of an electron pair) [20]. Shifting of the Fermi level towards the conduction band is another effect commonly associated to the N-doping of semiconductor carbon matrixes. According to some studies [21], nitrogenation of carbon matrixes may also influence structural parameters such as pore size distribution and surface area. Usually N atoms substituted to carbon matrixes improve the pseudocapacitance of electrochemical energy storage devices [22].

Thus, facile and high-yield methods are explored for N insertion to carbon matrixes (especially graphene planes) since graphene’s chemical reactivity is limited. On the contrary to graphene oxidation, which is relatively easy to perform and has been known for a long time by standard wet chemistry oxidants like KMnO_3_ [23], the nitrogenation of graphene requires a more complex procedure and even sophisticated instruments. Several methods for the nitrogenation of graphite/graphene are commonly exploited [24,25,26] (Table 1).

The reviews on the synthesis of N-doped graphene covering the latest achievements, regardless of the place/date of publishing and the number of cited references, do not reveal any other nitrogen enrichment procedures except the “classical” ones (Table 1). The methods suffer from plenty of technological disadvantages, among which very low or low yield seems to be the main obstacle refraining industrial up scaling. Some methods rely on graphene oxide (GO) instead of pure graphene, which additionally limits their applicability since GO is another material that is different from graphene, i.e., GO has, for example, an inferior electric conductivity, which is a key property to be exploited when such materials are utilized for electrode design in electrochemical devices. To date, hundreds of authors whose papers have already been reviewed have mentioned the idea of nuclear radiation as factor triggering a direct reaction between graphene/graphite and a nitrogen source being in contact with the carbon matrixes. In fact, some publications have announced the applicability of gamma radiation to initiate a reaction between some solids and chemical modifiers (including graphene/graphite and their derivatives) as presented in Table 2. Even when citing more references, no trace can be found on the application of gamma radiation for triggering the reaction between graphene/graphite and gaseous/liquid nitrogen carries. In our opinion, there is no predecessor to the current study, of which the aim is to investigate the qualitative and quantitative effects of gamma-induced nitrification of graphene. The results will be compared to traditional nitrification findings (Table 1).

Heteroatom insertion in the case of graphene is particularly difficult due to a relatively high chemical energy. The process is much easier in the case of carbon materials of lower crystallinity (partly amorphous) like carbon black, pyrolytic chars and activated carbons, in which heteroatoms may appear due to their transfer from a source material upon carbonization. Another way of heteroatom insertion is based on the presence of some surface reactive centers on amorphous carbon surfaces (free radical, unsaturated valences etc.). Graphene hardly undergoes such heteroatom modification scenarios. Therefore, any method enabling the insertion of heteroatoms has a practical meaning. This fact points out the importance of the research concept presented in this paper.

## 2. Results and Discussion

The morphology details of representative materials and pristine graphene nanoplatelets are shown in the SEM and HRTEM images (Figure 1). Figure 1a,e present the SEM images of GF_750_D_1D sample and GF_750_W_3D sample (please see the materials and methods section). The SEM images show several thin sheets. In particular, in Figure 1e, changes are clearly visible on the surface after the adsorption process using the wet method. Figure 1b,c,f present HRTEM images. In Figure 1b, a few layers of superimposed graphene (white arrow) are visible. Figure 1c shows the tinny and pure graphene nanoplatelets before the nitrogen introduction process. The graphene layers are clearly visible for all analyzed samples. The gamma irradiation did not destroy the original morphology of graphene nanoplatelets (the presented images are representatives for all performed tests). In turn, scanning electron microscopy with energy dispersive X-ray analysis (SEM/EDX) mapping images (Figure 1d,g) show the coverage of all carbon material with nitrogen, regardless of the used method. The analysis of all images (Figure 1) delivered direct proof of nitrogen presence and leads to the conclusion that the new proposed methods are highly effective to produce N-doped graphene.

The elemental composition for the N-doped graphene samples are given in Table 3. In general, nitrogen can positively influence the electric conductivity of graphene. Table 3 proves that the starting material, i.e., commercial graphene nanoplatelets is itself the source of small amount of nitrogen (below 0.8 wt.%). The elemental analysis revealed that the content of nitrogen in the proposed methods is up to 4 wt.%. In the dry method, when we increased the dose rate the nitrogen content decreased. The “dry method” was more effective regarding the intensity of N-doping, i.e., 4.19% of N was achieved for “dry method” in comparison to 2.00% of N for “wet method” (750 nanoplatelets). Moreover, the high content of 4.19% N was noted for a lower irradiation does collected in a shorter irradiation time of 14 days. The investigated N-doping methods “wet” and “dry” are characteristic because of saturation effect, i.e., N-content becomes stable or even starts to decrease after crossing a certain irradiation dose/irradiation time. Thus, in the “wet method” N-content increased constantly in order 1D –> 2D –> 3D for platelets of all types. In the “dry method”, the highest N-content was reached just after the dose 1D was accepted by the graphene samples. Then, the N-content remained stable or slightly decreased upon doses 2D and 3D.

Graphene nanoplatelets of the 750 kind are the most susceptible to N-doping by gamma radiation despite of the type of doping approach (“wet” or “dry”). The value 750 is the level of average surface area, which in this case is close to 750 m^2^/g. The high surface area is achieved due to the diminishing size of graphene plates (single and π-π stacked). Size diminishing means an increase in reactive centers located at the edges of graphene sheets. For smaller sheet the ratio: number of edge atom-to-mass is higher and therefore better for processes exploiting the reactivity of outer carbon atoms. Due to the same reasoning, the intensity of N-doping for the 150 nanoplatelets series (150 m^2^/g) is less spectacular. Elemental composition regarding C content in pristine (raw) nanoplatelets is inconsistent with their surface area. Lower C-content of 150/750 series in comparison to 300 series is a sign of the natural reactivity of the material exposed to air/oxygen even at room temperature. The highest C-content for 300 raw nanoplatelets is expectedly accompanied by the lowest O-content bonded to graphene planes as a result of a spontaneous reaction between carbon (graphene) and oxygen atoms (air).

Thus, the 300 platelets are the less reactive material among all graphene platelets under investigation. The assumed highest chemical stability for 300 nanoplatelets resulted in the lowest susceptibility to N-doping even in the experimental conditions conducive to chemical reaction (gamma irradiation).

The “wet” and “dry” methods differ because of the triggering of co-reactivity to oxygen. In both methods, oxygen is present in the reaction space: atmospheric oxygen (“dry” method) and water included oxygen (“wet” method). The mentioned difference applies to the case of the most reactive material, i.e., 750 nanoplatelets treated with 3D dose, where the oxygen content increased to the range of 12.395–14.72% O (“wet” method) from 11.09% O for pristine material. A similar phenomenon did not occur for “dry” method, when a slight decrease in O-content might be observed upon doses 1D, 2D and 3D.

All gamma radiation-treated samples are characteristic because of very high C-content, which is definitely better than that for graphene oxide (C/O 1.5–2.5). The C-to-O mass ratio is differentiated regarding the doping method and the nanoplatelets kind.

Raw graphene: 12—150 platelets, 85—300 platelets, 8—750 platelets.

“Dry” method: 16–85—150 platelets, 18–95—300 platelets, 8–9—300 platelets.

“Wet” method: 12–14—150 platelets, 13–45—300 platelets, 6–7—750 platelets.

C/O ratios for gamma irradiated 750 nanoplatelets (“dry” method) are on average similar to C/O ratios measured for pristine graphenes. C/O ratios for “wet” method modified platelets are the lower than for raw graphenes and lower than corresponding values for materials subjected to “dry” treatment. Thus, the “wet” method beside N-insertion doubtlessly leads to oxidation of graphene matrixes as a side (unwanted) effect.

The XRD diffractograms of the obtained materials are shown in Figure 2. Sharp peaks in samples of the GF_150 and GF_300 series (Figure 2a,b) indicate an ordered and crystalline structure. In turn, for the GF_750 series (Figure 2c), the amorphous content is higher because the peaks are not so sharp. The structure of the used starting material affects the appearance of such XRD diffractograms. The crystal plane diffraction peak of the obtained materials appears at ~23° (002 reflection of graphitic carbon) and a smaller peak at ~44° (100 reflection of graphitic carbon) was determined.

The Raman spectra (532 nm laser) of the obtained materials are shown in Figure 3. In graphite, a single sharp G peak is present at 1582 cm^−1^ and corresponds to the tangential vibrations of carbon atoms. Thus, G peak intensity may be a good measure of a sample’s graphitization. N-doped graphene exhibits two strong characteristic bands at 1340 cm^−1^ (D band) and 1580 cm^−1^ (G band). These bands are characteristic for graphene [38,43]. In turn, the 2D band is a characteristic peak of the graphene structure.

The ratio of the intensities between the D and G bands (I_D_/I_G_) depends on the level of the disorder. The I_D_/I_G_ ratios for GF_150 series are between 0.23 and 0.50, for GF_300 series are between 0.45 and 0.70, and for GF_750 series are between 0.58 and 0.94, respectively, as shown in Figure 4. The increase in the I_D_/I_G_ ratio clearly demonstrates the defects in the graphene structure. The differences in the I_D_/I_G_ ratios for three obtained series confirm that the materials contain a different number of defects in the carbon structure.

According to Figure 4, the most defective structure with a higher degree of graphitization is exhibited by samples from the GF_750 series. The observed ratios of N-doped graphene in the second method are almost similar. In contrast, the I_2D_/I_G_ ratio for all investigated series is in similar range between 0.40 and 0.50. It is known that the ratio I_2D_/I_G_ of the intensity is dependent on the number of graphene layers. This ratio means that the obtained N-doped graphene structures are multilayers.

The results of a combustion elemental analysis allowed the determination of the overall bulk content of three principal elements: C, N and H, while the specific chemical environment of these elements and oxygen remains unknown. Therefore, the last conclusion is supported by X-ray photoelectron spectroscopy (XPS) investigations. Typical N-insertion methods reviewed in the introduction are characteristic because of diversification of N atoms bonding to graphene matrixes. Several structures are typically assumed: quaternary, pyridine, pyridone, amine, pyroll, imid, amid, lactam, among which quaternary groups are of primary importance for electrode-based applications (batteries, supercapacitors). XPS studies were aimed at the detection of the chemical environment of inserted N atoms.

The XPS spectra of the two representative samples of investigated series were determined and are demonstrated in Figure 5 and Table 4. The analysis of C1s energy is essential in determining the chemical bonding of carbon atoms. The elemental content of carbon was high, ranking from 92.2 to 96.1 at.% in samples of the wet method and from 90.9 to 96.6 at.% in samples of the dry method. Carbon atoms were mostly bonded as sp^2^ hybridized atoms (band C 1s at binding energy 284.6 eV), which is characteristic for graphene materials. The C1s spectra of N-doped graphene are composed of seven peaks corresponding to: C=C bond (sp^2^) peak at 284.6 eV [44], C–C bond (sp^3^) peak at 285 eV [44,45], C–O–C or C–OH or C–NH bond peak at 286.3 eV [40,41], C=O or O–C–O or N–C–O bond peak at 287.7 eV [44,45], O–C=O peak at 288.6 eV [44], and peaks with binding energy at 289.6 eV and 292.1 eV are associated with shake-up excitation [46]. The total amount of oxygen in samples of the wet method is in the range from 2.5 to 5.0 at.% and 2.4 to 5.3 at.% in the samples of the dry method, respectively. The peak at 532.0 eV signifies the presence of a O=C–N or C=O bond and the peak at 533.3 eV is characteristic of a O*=C–O or O–C–O bond [44,45,47]. In turn, the elemental content of nitrogen is in the range from 1.0 to 3.8 at.% for all proposed methods. The high-resolution N1s spectra can be deconvoluted into one peak, located at 400.5 eV, which is attributed to quaternary (N–Q) groups [44,45], respectively. The essence of the quaternary group is a cation consisting of a central positively charged atom with four substituents. It is known that the N–Q groups have a stronger donor electron character. This ability is important for electrochemical studies for the electron transfer, especially for the construction of electrodes in solar cells, supercapacitors or metal–air batteries [48,49,50,51,52,53].

Sliwak et al. [53] reported that a low oxygen content and the presence of stable quaternary nitrogen groups improved the charge propagation on the carbons. The presence of such an N–Q bond can facilitate the electron transfer and cause the enhancement of conductivity in carbons. XPS results confirmed the successive introduction of a high nitrogen content into the graphene structure. Regardless of the used methods, N-doped graphene containing nitrogen at a similar level was obtained.

Gamma irradiation causes the formation of reactive sites in the carbon matrix of an ionic and/or radical nature. In the obtained systems, the carbon matrix is directly enriched. This means that the enrichment of graphene in heteroelements can be obtained after the production of a porous graphene matrix. The proposed methods are characterized by a negligible consumption of reagents and a negligible amount of waste.

## 3. Materials and Methods

N-doped graphene was synthesized in two ways. In the first method called the “dry method”, 1g of graphene nanoplatelets (three different series, MO purchased from Sigma Aldrich were added to plastic test tubes and left open. The ethylamine (St. Louis, MO, USA) purchased from Sigma Aldrich was opened in a container containing ice. This was caused by the fact that the boiling point of this substance is 17 °C. Next, 10 mL of ethylamine was placed in a separate and open test tube. All test tubes were sealed in a square transparent plastic box. Then, we inserted the box into a radiation device. In the second method called the “wet method”, 1 g of graphene nanoplatelets (three different series) purchased from Sigma Aldrich were added to 100 mL PVC containers (Bionovo, Legnica, Poland). Next, we added 25 mL 6M NH_3_ aq (POCH, Gliwice, Poland) to each container and the contents were well mixed, and the containers were closed. Then, we inserted the containers into a radiation device.

Irradiation was carried out in a radiation device RChM-γ-20, where the source of gamma radiation is Co-60. The dosing rate during the experiments ranged from 19.3 to 21.0 Gy/h, average value ~20 Gy/h, determined using a Fricke dosimeter. Periodically, further samples were taken for analysis in a specified time schedule. After each irradiation step, the material was washed repeatedly with distilled water (2 L) using a Büchner funnel (Bionovo, Legnica, Poland) to remove all impurities, in particular, the residues of ethylamine and ammonia. Then, it was dried in an electric furnace at 150 °C for 24 h.

The N-doped graphene samples obtained by the proposed methods were denoted as GF_X_Y_ZD, where: e.g., GF_X means the type of used raw material (150—graphene nanoplatelets with an surface area of 150 m^2^/g; 300—graphene nanoplatelets with an surface area of 300 m^2^/g; 750—graphene nanoplatelets with an surface area of 750 m^2^/g), Y means the method used (D—dry method; W—wet method), and ZD means the dosing rate for dry method (1D—7.6 kGy; 2D—14.3 kGy; 3D—27.5 kGy) and for wet method (1D—5.5 kGy; 2D—14.6 kGy; 3D—20.9 kGy). GF_X_raw—means raw material before processing.

The morphology of the N-doped graphene was analyzed by scanning electron microscopy (SEM, 1430 VP, LEO Electron Microscopy Ltd., Oberkochen, Germany). The obtained materials were also examined by high-resolution transmission electron microscopy (HRTEM, FEI Europe production, model Tecnai F20 X-Twin, Brno, Czech Republic). The carbon materials obtained prior to the HRTEM microscopic analysis were dispersed in ethanol and treated with an Inter Sonic IS-1K bath for 15 min and deposited on holey carbon-coated copper grids. The volumetric elemental composition (carbon, nitrogen and hydrogen) of the materials was analyzed by means of a combustion elemental analyzer (Vario MACRO CHN, Elementar Analysensysteme GmbH, Langenselbold, Germany). Raman spectra were obtained by a micro-Raman spectrometer (laser wavelength 532 nm, Senterra, Bruker Optik, Billerica, MA, USA). The laser was tightly focused on the sample surface through a 50× microscope objective. To prevent any damage of the sample, excitation power was fixed at 2 mW. The resolution was 4 cm^−1^, CCD temperature 223 K, laser spot diameter 2.0 µm, and total integration time 100 s (50 × 2 s), were used. X-ray diffraction (XRD) patterns of the samples were obtained with an X-ray diffractometer (X’Pert-Pro, Philips, Cambridge, UK) equipped with a Cu Kα source at 40 kV and 30 mA. X-ray photoelectron spectroscopy (XPS, PHI5000 VersaProbe II Scanning XPS Microprobe, Chigasaki, Japan) measurements were performed using a monochromatic Al Kα x-ray source. Survey spectra were recorded for all samples in the energy range of 0 to 1300 eV with a 0.5 eV step, high-resolution spectra were recorded with a 0.1 eV step.

## 4. Conclusions

In summary, we have demonstrated new alternative methods and a low-cost strategy for obtaining N-rich graphene. The N-doped graphene was successfully prepared using gamma irradiation at a radiation dose lower than 30 kGy. The most efficient method is the “dry method” with the use of ethylamine, where the nitrogen content exceeds 4 wt.%. The irradiation method is very selective regarding the chemical form of inserted N atoms, which are exclusively bonded as quaternary groups, i.e., functional groups exceptionally useful for the improvement of such doped graphene electrodes in batteries and supercapacitors. On the contrary, traditional N-doping methods yield very diversified forms of N bonding, among which only few are expected to be used in the mentioned electrochemical applications. The proposed methods are a simple and universal option to the already known methods of introducing nitrogen and other heteroatoms, such as sulfur, boron or phosphorus.

This aspect of the performed study will be continued in further research. The future work should concentrate on two basic directions.

Firstly, the influence of raw graphene structure on the qualitative and quantitative effectiveness of N-enrichment should be investigated. In this study, mostly MLG was investigated which meant that any of the applied N-bearing reagents had limited access to the reactive centers induced by gamma radiation in the intrinsic spaces inside the MLG granules. The authors assume that there might be a substantial difference whether SLG, FLG or MLG is subjected to the nitrogen enrichment according to the radiation procedures presented in this study. The less agglomerated graphene precursors like SLG or FLG should react intensively, yielding graphene-based materials of a higher N content than in the case of MLG. For this purpose, several graphene splitting methods seem to applicable like electroexfoliation combined with the wet-chemistry exfoliation by means of polar aprotic liquids (acetone, NMP etc.). Intensively electroexfoliated graphene will be investigated as a continuation of the current study, which can serve as a reference point.

Secondly, the discovered unified chemical bonding of nitrogen to graphene matrix should result in the advancement of the electrochemical properties of them. Therefore, such gamma radiation-modified graphene-based materials should be evaluated electrochemically as an electrode material in some batteries such as metal–air ones. Oxygen reduction reaction (ORR) will be investigated aiming at the determination of cathodic properties of the materials.

Thirdly, there are neither theoretical nor experimental obstacles against nitrogen enrichment of other carbon-based materials like carbon nanotubes and even ordinary activated carbons. Nitrogen-rich activated carbons are still an emerging field in the search for effective electrode materials for plenty of electrochemical devices like metal–air batteries, supercapacitors, and fuel cells.

## Figures and Tables

**Figure 1 materials-13-04975-f001:**
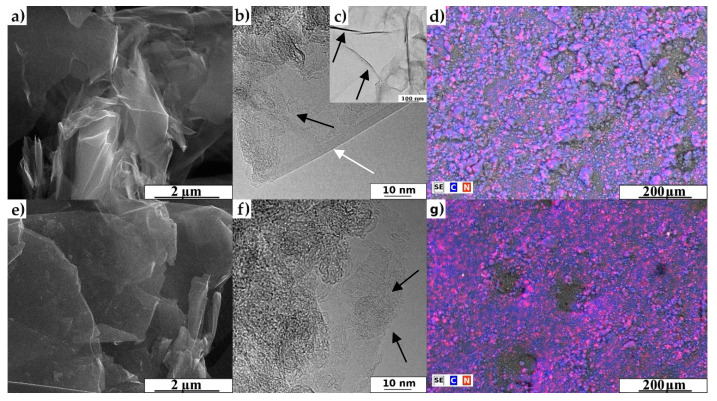
Representative images of obtained samples: (**a**) SEM image of GF_750_D_1D sample, (**b**) HRTEM image of GF_750_D_1D sample, (**c**) HRTEM image of pristine GF_750_raw sample, (**d**) SEM/EDX mapping of GF_750_D_1D sample, (**e**) SEM image of GF_750_W_3D, (**f**) HRTEM image of GF_750_W_3D and (**g**) SEM/EDX mapping of GF_750_W_3D sample (arrows on the image 1c show pure and thin sheets).

**Figure 2 materials-13-04975-f002:**
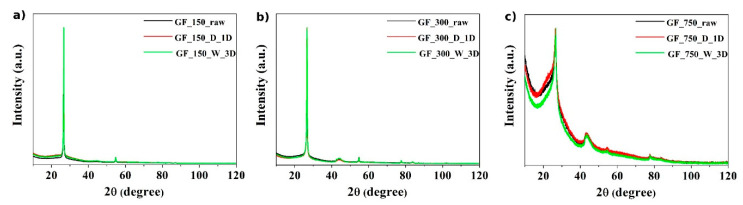
XRD images of three different series of obtained samples: (**a**) GF_150 series, (**b**) GF_300 series and (**c**) GF_750 series.

**Figure 3 materials-13-04975-f003:**
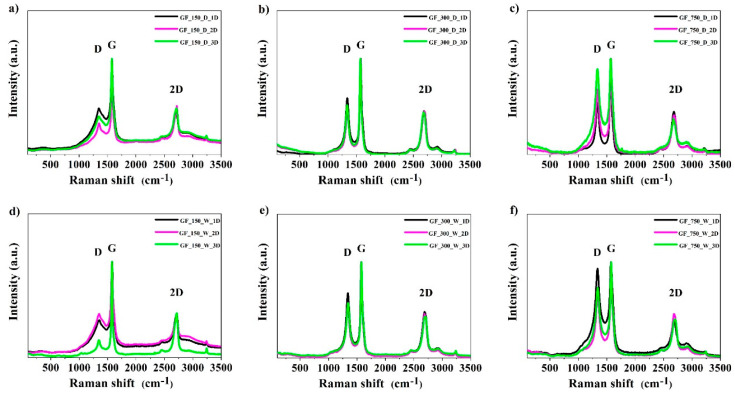
Raman spectra of all investigated series of the obtained samples: (**a**) GF_150_D series, (**b**) GF_300_D series, (**c**) GF_750_D_series, (**d**) GF_150_W series, (**e**) GF_300_W series and (**f**) GF_750_W series.

**Figure 4 materials-13-04975-f004:**
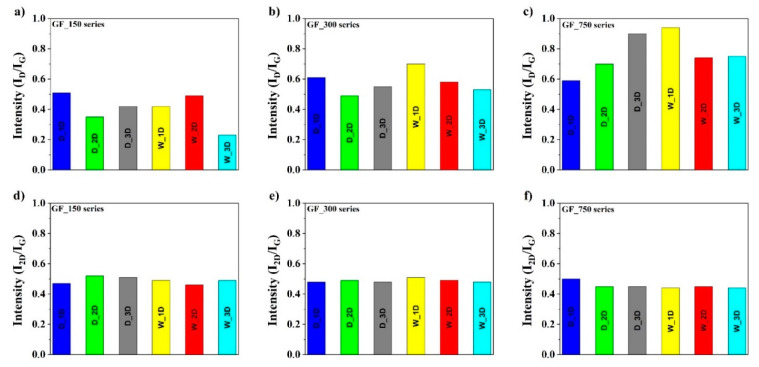
The ratio of the intensities between the D, G and 2D bands: (**a**) I_D_/I_G_ of GF_150 D and W series, (**b**) I_D_/I_G_ of GF_300 D and W series, (**c**) I_D_/I_G_ of GF_750 D and W series, (**d**) I_2D_/I_G_ of GF_150 D and W series, (**e**) I_2D_/I_G_ of GF_300 D and W series and (**f**) I_2D_/I_G_ of GF_750 D and W series.

**Figure 5 materials-13-04975-f005:**
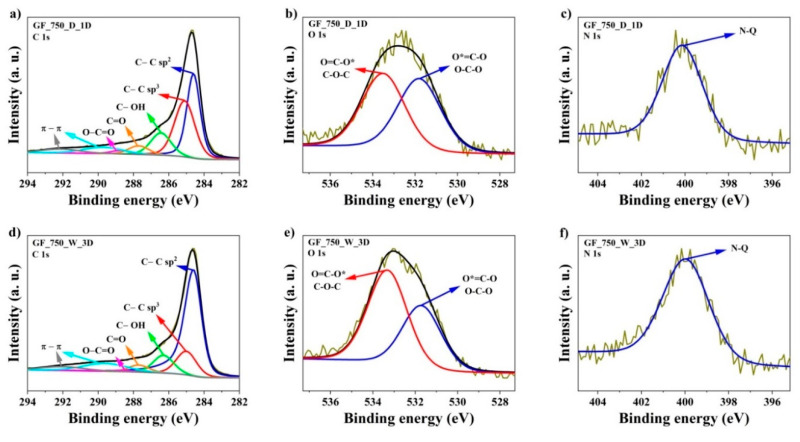
High-resolution X-ray photoelectron spectra for C1s, N1s, and O1s of representative samples: (**a**) C1s of GF_750_D_1D, (**b**) O1s of GF_750_D_1D, (**c**) N1s of GF_750_D_1D, (**d**) C1s of GF_750_W_3D, (**e**) O1s of GF_750_W_3D and (**f**) N1s of GF_750_W_3D.

**Table 1 materials-13-04975-t001:** An overview of currently practiced methods for nitrogen insertion into graphene.

Method	Description	Drawbacks	Example Study
CVD	High temperature furnace up to 1000 °C, vacuum 1 Torr, catalyst, NH_3_ as nitrogen source, He as shielding gas	Complex instrumentation, very low yield	[27,28]
Arc Discharge	Electric arc discharge conditions, pyridine and ammonia as a nitrogen carrier	Complex instrumentation, difficult to control, very low yield	[29]
Pyrolysis	High temperature pyrolysis of a stolid mixture of GO—urea lattice, respectively	Limited yield, long time high temperature synthesis, application of GO instead of pure graphene	[30]
Heat treating	Heating to 800–1000 °C a solid mixture of GO-nitrogen source, neutral atmosphere, melamine as a potential nitrogen source		[31]
Solvothermal	200–300 °C, 4–5 h duration, dimethylformamide as a solvent and nitrogen source	Yield limited by the experimental vessel volume, use environmentally and health unfriendly reagents	[32]
Gas Annealing	High temperature of 500–1000 °C during electrical annealing of GO in nitrogen atmosphere, ammonia gas (NH_3_) as a nitrogen source	GO applied instead of pure graphene/graphite, low yield, a high temperature method	[33,34]
N_2_ Plasma Treatment	Nitrogen content controlled by the plasma strength and exposure time, example plasma generator parameters 40–200 W, 900 V DC bias, high vacuum 200 mTorr, 20–80 min treatment, graphene or GO as a key precursor, N_2_ and NH_3_ as nitrogen source	Sophisticated instrumentation and challenging synthesis conditions, low yield	[35]
Dry Ball Milling	Mechanochemical process, room temperature direct grinding of dry powdered graphite in the N_2_ or NH_3_ atmosphere, nitrogen content controllable by changing milling parameters	Unwanted insertion of impurities from the grinding setup, which must be removed by additional treatment, laboratory scale process	[36]
Nanoscale High Energy Wet Ball Milling	Mechanochemical process, room to 80 °C wet milling; gas, liquid and solid nitrogen carriers permittable, GO advised as carbon precursor	Complex manufacturing pathway including frequent rising, laboratory scale process	[37]

**Table 2 materials-13-04975-t002:** Example application of gamma radiation for physical and chemical modification of carbon-type materials.

Experiment Description	Example Study
Gamma-ray radiation for altering of physicochemical properties of graphene oxide (GO) by insertion of aminosilanes	[38]
Metal-to-insulator transition of monolayered graphene	[39]
Changing activated carbon surface upon gamma irradiation in the presence of pure and contaminated water	[40]
The effect of gamma irradiation on the structure and composition of chemically synthesized few-layered graphene-based materials	[41]
Surface activation of commercial activated carbon induced by gamma radiation and the applicability of modified carbon for the removal of odors	[42]

**Table 3 materials-13-04975-t003:** Elemental composition of obtained N-doped graphene.

Method	Sample	Elemental Analysis (wt.%)			
		N	C	H	Residuals (Oxygen)
	GF_150_raw	0.61	90.77	0.90	7.72
	GF_300_raw	0.34	98.04	0.47	1.15
	GF_750_raw	0.72	87.32	0.87	11.09
dry	GF_150_D_1D	2.19	90.79	1.55	5.47
	GF_150_D_2D	2.14	91.88	1.63	4.35
	GF_150_D_3D	2.09	94.92	1.83	1.16
	GF_300_D_1D	1.66	93.68	0.88	3.78
	GF_300_D_2D	1.66	92.44	0.85	5.05
	GF_300_D_3D	1.66	96.35	0.96	1.03
	GF_750_D_1D	4.19	83.95	1.64	10.22
	GF_750_D_2D	4.03	84.66	1.55	9.76
	GF_750_D_3D	4.01	83.42	1.71	10.86
wet	GF_150_W_1D	0.62	90.52	1.15	7.71
	GF_150_W_2D	1.29	90.77	1.36	6.58
	GF_150_W_3D	1.38	89.84	1.11	7.67
	GF_300_W_1D	0.77	91.85	0.68	6.70
	GF_300_W_2D	0.98	95.70	0.72	2.60
	GF_300_W_3D	1.17	96.13	0.54	2.16
	GF_750_W_1D	1.52	82.85	0.91	14.72
	GF_750_W_2D	1.98	84.12	1.51	12.39
	GF_750_W_3D	2.00	84.27	1.23	12.50

**Table 4 materials-13-04975-t004:** Elemental composition of the obtained N-doped graphene from the XPS spectra.

Element	C								O			N
Binding Energy (eV)	284.6	285.0	286.3	287.7	288.6	289.6	292.1	% of Total	532.0	533.3	% of Total	400.5
Sample	Content (at.%)								Content (at.%)			Content (at.%)
GF_150_D_1D	52.8	23.9	7.4	2.8	1.2	5.1	3.4	96.6	1.0	1.4	2.4	1.0
GF_150_W_3D	60.8	13.9	6.5	2.7	0.3	6.7	4.8	95,7	1.6	1.5	3.1	1.2
GF_300_D_1D	45.5	23.2	9.8	3.8	2.0	6.7	4.5	95.5	1.6	1.5	3.1	1.4
GF_300_W_3D	54.1	17.3	7.8	3.5	0.1	8.0	5.3	96.1	1.0	1.5	2.5	1.4
GF_750_D_1D	28.3	34.5	10.8	4.8	3.2	5.6	3.7	90.9	2.8	2.5	5.3	3.8
GF_750_W_3D	52.8	12.4	9.3	3.7	0.8	8.9	4.3	92.2	2.0	3.0	5.0	2.8

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
