# Peer review of "Highly Effective Methods of Obtaining N-Doped Graphene by Gamma Irradiation"

_materials, 2020, doi:10.3390/ma13214975_

Round 1
Reviewer 1 Report
We have read the manuscript titled “Highly effective methods of obtaining N-doped graphene by gamma irradiation”. The authors proposed a new strategy of gamma irradiation to synthesis of N doped graphene. The method is novel and of scientific significance, and the results are distinct and organized well. Therefore I think this manuscript is acceptable for publishing in Materials. In addition, two small questions need to be solved.
Firstly, In the introduction part, the authors mainly focused on the synthesis of N doped materials, while the purpose of N doping should be supplied. Generally speaking, N doping is a common strategy to endowing new properties into a material. Such as improving conductivity ( J. Mater. Chem. A, 2019, 7, 4729; Nano-Micro Lett., 2020, 12, 20 ) constructing new bonds with other atoms (Appl. Phys. Lett. 2020, 116, 113102). So I think the importance of N dopants should be illustrated in this part.
Besides, The elemental compositions are estimated as wt.% (Table 3). What method did the authors use to obtain these compositions? These should be emphasized. Besides, I think the fractions by weight (wt.%) is not visible enough. In my opinion, it is better to use molar ratios to illustrate the elemental composition.
Reviewer 2 Report
In this manuscript, Piotr Kamedulski et al. have shown an interesting method to fabricate nitrogen-doped graphene with a maximum nitrogen content of 4 wt.% and reported the physical and chemical properties of these materials utilizing SEM, HRTEM, XRD, XPS. I find this article intriguing and suitable to publish in the Materials journal. But before the publication, they should address the following questions/comments:
- The authors should rephrase this sentence: "Theoretically pristine graphene ……..C atoms of each graphene plane. It is not clear to the readers.
- In the following sentence, few-layered graphene and multi-layered graphene should be written with small letters as single-layer graphene or use capital letters for all of those names for the consistency. Also, please rephrase the last part of the sentence "however, those application……..also high":
"Instead of SLG other forms of graphene are privileged as Few Layered Graphene (FLG) and Multi-Layered Graphene (MLG); however, those application potential is also high."
- How do graphene's physical/chemical/electronic properties differ with the doping of other heteroatoms (such as sulfur, boron, phosphorous, oxygen)? Why are the authors interested in the doping of nitrogen only? If there is any particular advantage of nitrogen doping compared to others should be discussed. What about the co-doping of heteroatoms? The authors should discuss this in the main text.
- The authors have introduced the name of the samples (GF_750_D_1D and GF_750_W_3D) in a short form at the beginning of the results and discussion section without any explanation. They should at least mention that "please see the method section" which would help the readers.
- The authors should report the AFM data to have a better idea about the presence of a number of graphene layers.
- For the wet processing technique, have the authors tried to irradiate the sample for more than 3D to see how the trend goes? Will the saturation effect take place?
- In this manuscript, the authors only showed the change in the physical and chemical properties of graphene upon the doping of nitrogen, but they have not explored the variations in electronic properties. They should consider reporting the electronic properties.
- The conclusion could be more critical in order to explain the structure-properties relationships.
- The authors should discuss clearly about the novelty of this work. Also, they should add some discussion about future work.
Author Response
Please see the attachment

This manuscript is a resubmission of an earlier submission. The following is a list of the peer review reports and author responses from that submission.